# *Francisella* and Antibodies

**DOI:** 10.3390/microorganisms9102136

**Published:** 2021-10-12

**Authors:** Klara Kubelkova, Ales Macela

**Affiliations:** Department of Molecular Pathology and Biology, Faculty of Military Health Sciences, University of Defence, 1575 Terebesska, 500 01 Hradec Kralove, Czech Republic; ales.macela@unob.cz

**Keywords:** natural immunity, natural antibodies, B cells, *Francisella tularensis*, intracellular pathogen

## Abstract

Immune responses to intracellular pathogens depend largely upon the activation of T helper type 1-dependent mechanisms. The contribution of B cells to establishing protective immunity has long been underestimated. *Francisella tularensis*, including a number of subspecies, provides a suitable model for the study of immune responses against intracellular bacterial pathogens. We previously demonstrated that *Francisella* infects B cells and activates B-cell subtypes to produce a number of cytokines and express the activation markers. Recently, we documented the early production of natural antibodies as a consequence of *Francisella* infection in mice. Here, we summarize current knowledge on the innate and acquired humoral immune responses initiated by *Francisella* infection and their relationships with the immune defense systems.

## 1. Introduction

In approaching the subject of acquired humoral immune responses initiated by *Francisella* infection and their relationships with the immune defense systems, we take as our point of departure the June 2007 review article on this subject published by Elkins et al. Those authors exhaustively summarized the current state of knowledge at that time regarding the relationships between *Francisella* microorganisms and the host immune system as obtained from animal models and the study of human infections up to their convalescence [1]. Dozens of reviews covering various themes of induction, regulation, and expression of the host’s immune responses to *Francisella* at molecular and cellular levels have been published since that time. Thus, we have today general knowledge regarding innate immune recognition [2,3,4,5,6] and the engagement of neutrophils [7,8,9], macrophages [10,11,12,13], and dendritic cells [14,15,16,17] during the phase of immune response expression. We nevertheless continue to have substantial gaps in understanding the processes during *Francisella*–host immune system interaction, wherein B cells and antibodies [18,19,20,21] might play significant roles during innate immune response, just as in the adaptive phase of immune response to *Francisella* microbes.

Important characteristics of *Francisella* species include their abilities to infect a broad range of organisms, ranging from amoebas [22], ticks [23], mosquitoes [24], fish [25], amphibians [26], and birds [27,28] to diverse mammals, including rodents, lagomorphs, carnivores [29], monkeys [30], and humans [31], and to accomplish their replication cycle in a diverse assortment of eukaryotic cells. The *Francisella* virulence factors, the intracellular lifestyle of *Francisella*, and its interaction with individual cell organelles are not yet sufficiently understood. Similarly, difficulties in characterizing the molecular and cellular defense responses of an infected organism against *Francisella* are due to the characteristic behavior of *Francisella*, which weakly activates cells of the mammalian innate immune system and actively suppresses host cell responses [32]. In particular, consequences of the relationships among *Francisella*, B cells, and natural and actively induced antibodies are still debated. Here, we summarize recent knowledge on this issue and present our view as to the role of antibodies during the interaction of *Francisella* with the host’s cellular and molecular system of defense. The data regarding the antibodies against *Francisella* originated from ecological and epidemiological studies, clinical data, experimental studies oriented upon the *Francisella*–host immune system interactions, and studies devoted to the development of tularemia vaccines.

## 2. Interactions of Hosts with *Francisella* in Nature Leave Significant Antibody Traces

Because tularemia is widespread throughout the northern hemisphere, most studies came from Europe, Asia, and North America. Substantial numbers of wild and domestic animals, as well as humans, living in endemic foci of tularemia have serum antibodies against *F. tularensis*. Seroprevalence of tularemia ranges from a few tenths of a percent to tens of percentage points, depending upon the area and season of screening. European wild small mammals (*Apodemus flavicollis*, *Myodes glareolus*, *Sorex araneus*, *Apodemus sylvaticus*, *Apodemus agrarius*, *Microtus arvalis*, and one *Talpa europaea*) trapped at three localities in the Czech Republic were found to have antibodies against *F. tularensis*, and the prevalence of antibodies was significantly different among animal species and sex [33]. Thirty-four blood samples from 656 Swedish wild predators and scavengers, among them brown bear (*Ursus arctos*), Eurasian lynx (Lynx lynx), raccoon dog (*Nyctereutes procyonoides*), red fox (*Vulpes vulpes*), wild boar (*Sus scrofa*), wolf (*Canis lupus*), and wolverine (*Gulo gulo*), had antibodies against *F. tularensis* subsp. *holarctica* [34]. Substantial numbers of wild foxes (*V. vulpes*), raccoon dogs (*N. procyonoides*), and wild boars (*S. scrofa*) collected in several areas of Germany, as well as hunting dogs, were positive for antibodies to *F. tularensis* [35,36,37]. Exposure to wild and domestic animals expressing antibodies to *F. tularensis* were identified as a risk factor for humans in European parts of Turkey [38,39]. Rodents studied in western Iran (Hamadan Province) and belonging to species of the Persian jird (*Meriones persicus*) and Libyan jird (*Meriones libycus*) were tularemia-seropositive and showed no cross-reactivity with brucellosis [40]. Some Japanese wild animals, including black bears, were shown to be seropositive to *F. tularensis* antibodies [41,42]. Seroprevalence to tularemia of wild animals has also been demonstrated in various parts of Russia [43,44,45], Armenia [46], and North America. Antibodies to *Francisella* have been detected in hares in Ontario, Alberta, and Nova Scotia [47,48,49,50], snowshoe hares, muskrats, and coyotes (*Canis latrans*) in Québec [51], domestic dogs in New Mexico [52], prairie dogs in Texas [53], wildlife and humans in Alaska [54], and ground squirrels (*Spermophilus beecheyi*) in Oregon [55]. Generally known are studies on tularemia at Martha’s Vineyard in Massachusetts. Skunks and raccoons there were frequently seroreactive, whereas white-footed mice, cottontail rabbits, deer, rats, and dogs were not [56]. Landscapers mowing lawns on Martha’s Vineyard were also frequently seropositive [57].

Anti-*F. tularensis* antibody tests applied in epidemiological and ecological studies on tularemia have demonstrated spread of this etiological agent of tularemia, which is still considered to be a biological agent listed in category A, and these may serve as sentinels for tularemia prevention by identifying an area with increased risk of infection. In many cases, antibodies have been the only evidence of contact with the microbe because cultivation assays or PCR tests were negative. Moreover, individual antibody isotypes and their specificities, which are easily detectable and available in serum and saliva, are of particular importance for such purposes because they may also demonstrate the history of tularemia outbreak (infection’s early interval—IgM only, ongoing acute state—IgM and IgG, or infection occurred some time ago—IgG only) in the monitored area [51,58,59,60,61,62]. There exist a substantial number of tests and technologies that enable tularemia-specific antibody monitoring, and that can be utilized in natural environmental studies [63].

## 3. Humoral Immune Response to *Francisella* Infection and Vaccination

Most original sources in the literature state that the production of antibodies after natural infection with live *Francisella* culminates during the second week after infection. The IgM, IgG, and IgA isotypes generally occurred simultaneously when the agglutination test was used for antibody titer evaluation during the first week after infection. The IgG and IgA antibody isotypes could be detected earlier, however, using an enzyme-linked immunosorbent assay (ELISA) [64,65]. One later study further specified that, in serum samples obtained from patients with serologically confirmed tularemia, specific IgG2 antibodies predominated among the IgG isotypes. Specific antibodies of IgG1 and IgG3 but not IgG4 were also present [66]. Vaccination of human volunteers with a live-attenuated strain of *F. tularensis* induced a long-lasting humoral immune response, which could be reliably demonstrated the second week after vaccination and lasted minimally up to 18 months post vaccination [67]. The production of IgM, IgG, and IgA isotypes began simultaneously; all of these immunoglobulin classes were detectable up to 1.5 years after vaccination. Their response curves differed, however, with the agglutination titer of IgM Abs peaking earliest, the second month after infection, and the titer of IgG Abs peaking last, approximately in the fourth month post vaccination. Some data have demonstrated that the serum agglutinins against *Francisella* may persist for as long as 25 years after natural infection with *Francisella* [68]. However, the tests of cell-mediated immunity were more convincing than the agglutination test used for evaluation of humoral response [68], because cellular response persists longer than the antibody response. It was also demonstrated that the antigenic determinants participating in cell-mediated and humoral immune response are not identical. Determinants responsible for immunospecific lymphocyte stimulation were included into a group of proteins, whereas determinants inducing ELISA activity were mostly of a carbohydrate moiety [69]. In spite of the fact that *F. tularensis* lipopolysaccharide (LPS) is actually a weak immunogen and inducer of humoral immune response [70,71], human peripheral B cells, but not murine splenic B cells, produce detectable and progressively increasing IgM antibody levels over time [72]. This determination led to the notion that glycoproteins or proteins themselves, rather than LPS, may be the targets of antibodies. An immunoproteomic approach used to analyze the specificities of serum antibodies obtained from individuals who had contracted tularemia or from laboratory personal and clinical trial subjects who had been vaccinated with two types of tularemia vaccines, one of which was LPS, clearly identified the protein targets of anti-*F. tularensis* antibodies [73]. Antibodies recognized as many as 50 tularemia proteins, only 10 of which have no orthologs or analogs in eukaryotic cells [73]. An analogous study carried out using New Zeeland white rabbits immunized with a heat-killed *F. tularensis* live vaccine strain (LVS) revealed 28 *F. tularensis* LVS immunoreactive proteins, nine of which were not previously identified as immunoreactive by murine or human sera [74]. A majority of anti-*Francisella* antibody specificities were also identified in other human [64,75,76,77] and several experimental animal studies [78,79,80]. The numbers of anti-*F. tularensis* antibody specificities in these studies are not so important as is the fact that many of them are potentially reactive toward the host’s own protein targets.

Studies on diverse animal species infected by *F. tularensis* strains have revealed a uniformity of antibody production onset during the second week after infection or vaccination, as well as a significant dependency of antibody production intensity on the route of infection or vaccination. That routing may determine the type of the induced immune response and have bearing on the type of test that must be used to detect the antibodies [81,82,83,84,85]. The characteristics of *Francisella*-specific antibody production presented in the aforementioned studies were derived from samples collected during the adaptive immune response stage. However, the multireactive antibodies of IgM, IgG, and IgA isotypes are naturally present in normal human and animal sera and were originally known as autoantibodies (NAAbs). These NAAbs are produced by B1 B cells, encoded by germline genes with no, or few, somatic mutations, and they do not undergo affinity maturation in normal individuals [86,87]. The targets of these NAAbs are self-proteins, even heat stress proteins [88], complement components [89], and interleukins [90], carbohydrates, or sialoglycans [91,92]. These B1 cells can be found in the pleural and peritoneal cavities, while, in other tissues, such as lymph nodes, spleen, or bone marrow, the proportion of B1 cells is very limited [93]. The B1 cells in pleural and peritoneal cavities respond to infections by rapid activation, translocation to lymph tissues, and differentiation into antibody-secreting cells [94]. The whole transformation process of B1 cells into IgM plasma cells takes about 1–1.5 days [95]. Utilization of the bacterial *F. tularensis* infection model has revealed that appropriate antigenic stimulation, in this case by a glycolipid of *F. tularensis* LVS, induces IgM and IgG B1a antibody responses followed by the formation of long-lived T-independent antigen-specific B1a memory, both of which have markedly different responses from those of canonical B2 humoral immunity [96,97].

If the peritoneal B1 cells come into direct contact with *F. tularensis* strain FSC200, they respond immediately by producing natural antibodies, which we have provisionally termed infection-induced natural antibodies [98]. The lifespan of these antibodies in circulation seems to be very limited. Despite dissemination of the bacteria throughout the organs of infected mice, the spectrum of specificities has been observed to change during the first 48 h post infection. The targets for the majority of antibody clones were bacterial proteins having orthologs or analogs in the cytosol or mitochondria of eukaryotic cells [98]. Germ-free mice responded to *Francisella* infection by higher titers of all induced antibody isotypes and by a broader spectrum of antibody specificities than did specific-pathogen free mice. In both cases, mice responded dominantly to proteins phylogenetically conserved in both prokaryotic and eukaryotic cells. Thus, the quantity and the quality of such an early natural infection-induced antibody response are probably dependent on both phylogenetic and ontogenetic memory of the infected organism. An argument for the effect of phylogenetic memory can be founded upon the spectrum of antibody specificities induced by infection; an argument for ontogenetic memory can be based upon the long-term infection component-specific memory B1 cells arising from an immediate natural immune response [96,97,98]. The formation of memory cells specific for microbial components during primary natural immune response then favors the response of B1 cells to these targets during reinfection and, thus, limits the general spectrum of natural antibodies induced by primary infection. From an overall perspective, therefore, we can consider that rechallenge changes the preferences of self/nonself recognition. During primary response, the response to noninfectious self dominates over infectious nonself. This covers both self and microbial targets and ensures elimination of the dead cell remnants. Due to the presence of memory cells, the immune response to reinfection prefers the recognition of infectious nonself over noninfectious self. In this case, the limitation of microbial proliferation also limits the quantity of cellular debris that must be removed.

## 4. Role of Antibodies during Innate and Adaptive Phases of Immune Responses

The prevailing view that immunity to *Francisella* is mediated dominantly by T-cell-dependent immune responses has gradually changed since the beginning of the millennium. There are irrefutable data in the literature documenting the involvement of antibodies, as well as of B-cell subpopulations themselves, in processes, starting with primary interactions and up to the phase of secondary response to bacterial challenge.

### 4.1. Francisella and Serum Resistance: The Roles of Complement and Antibodies

The species of genus *Francisella*, as well as strains of *F. tularensis*, are to varying extents resistant to serum killing [99]. The knowledge concerning *Francisella* interaction with serum components remains nevertheless insufficient for understanding the nature of serum resistance. Several hypotheses have been put forward. It is generally accepted that the composition of *F. tularensis* LPS O antigen is a dominant factor enabling *F. tularensis* to resist both the complement attack and the cidal effects of other serum components. In spite of the different structures of *F. tularensis* subsp. *tularensis* and *F. novicida*, findings indicate that the O antigens of both *F. tularensis* subsp. *tularensis* and *F. novicida* play roles in protection from serum killing [100,101,102]. The role of the O antigen from *F. tularensis* subsp. *tularensis* appears to be relatively minor, however, compared to that of the *F. novicida* O antigen. This difference may indicate that, in *F. tularensis* subsp. *tularensis*, other surface structures or serum components play a role in resistance to serum killing [102]. One of these may consist of non-proteinaceous components of the outer membrane other than LPS transported by ValAB, which is a member of the superfamily of ABC transporters [103]. Furthermore, the deposition of factor H along with the iC3b fragment on the bacterial surface might be the principle for the serum resistance of *Francisella* [104]. Factor H is one of the potent soluble inhibitors of complement that, after binding onto target structures on cells or microbes, prevent complement activation [105]. Data in the literature have demonstrated that complement component fragments C3d, iC3b, and C4b are deposited onto the *F. tularensis* surface, which is a factor critical for classical pathway activation. In contrast, there are no data on deposition of the C5b–C9 membrane attack complex on the surface of *Francisella*. One of the explanations for why activation of the complete complement cascade by *Francisella* is incomplete and enables the *Francisella* to resist complement lysis can be seen in the association (binding) of factor H with the *Francisella* surface components. Nevertheless, the binding of factor H to the *F. tularensis* surface has not been sufficiently confirmed [106]. Moreover, the binding of some complement regulatory proteins may prevent complement activation on the surface of the bacterium. Binding of vitronectin, which is one of these proteins, may also be the mechanism via which *Francisella* strains evade complement attack [99].

The surface of *F. tularensis* binds the complement component fragments, but data on the requirement of C1q for optimal C3 deposition suggest the importance for IgM or IgG3 isotypes to initially bind on the bacterial surface [107]. The question, thus, may arise whether such a primary interaction of bacterium and complement components can ensue after binding of natural or cross-reactive antibodies on the bacterial surfaces. If such antibody specificities are not in circulation, the full-fledged complement activation may be absent, and that might be crucial for *Francisella’s* resistance to serum killing processes. Thus, a complex interaction of *Francisella* surface components with serum antibodies enabling the binding of complement components might constitute the first point of antibody engagement during mutual interaction of *Francisella* with the host’s defense system.

### 4.2. Initiation of Immune Responses

If the key to the binding of complement components to the surface of *F. tularensis* is indeed a consequence of previous interaction between the bacterium and antibody, then the antibodies already influence or control the bacterium’s primary interaction with the host cell. If only the actual primary host–pathogen interactions are considered, then isotypes of natural antibodies will control the nature of the infection. This opinion relates to the distribution of complement receptors (CRs) and Fc receptors of mammalian cell types. While macrophages and monocytes express complementary CR1, CR3, and CR4 receptors on their surfaces, follicular dendritic cells express CR1, CR2, and CR3 receptors, and B cells express only CR1 and CR2 receptors [108]. Similarly, Fc-receptor (FcR) types for IgG and for other switched Ig isotypes are dominantly expressed on phagocytic cells, including neutrophils and eosinophil granulocytes, but the FcμR binding IgM is selectively expressed only on T and B lymphocytes and natural killer (NK) cells in humans and exclusively on B cells in mice [109,110]. FcμR signaling is critical not only for B-cell survival and activation, IgM homeostasis, and regulation of humoral immune response, but also for resolution of infections [109,111,112,113].

Natural antibodies of IgM isotype are critical components during primary interaction of *Francisella* with human neutrophils and macrophages. Natural IgM binds to surface capsular and O antigen polysaccharides of *F. tularensis* and activates the classical complement cascade via C1q. That is followed by C3-opsonization of the bacterium. *Francisella* opsonized by C3 complement fragments is finally phagocytosed by human neutrophils via CR1 and CR3 acting in concert and by human monocyte-derived macrophages via CR3 and CR4 [114,115]. The IgM surface antigen receptor and CR1/2 of B cells are needed for *Francisella’s* internalization within the B cells [20]. Thus, natural IgM antibodies that opsonize the *Francisella* surface components initiate the classical complement cascade via C1q and promote internalization of *Francisella* through different complement receptors in a phagocyte-specific manner. Internalization of *Francisella* into the phagocytic cell is a prerequisite step for both dissemination and progression of infection on the one hand and antigen presentation to T lymphocytes on the other hand. The importance of fully functional B1a cells producing natural IgM antibodies has been demonstrated using μMT^−/−^ mice, which lack the μ chain in B cell development [116]. The μMT^−/−^ mice, having no mature B cells, exhibit increased susceptibility to primary infection with LVS, as well as reduced resistance to secondary infection and greater susceptibility to infection with virulent *F. tularensis* strain SchuS4, than do wild-type mice [117,118].

Thus, the IgM antibodies, as the first antibody isotype to appear during evolution, as well as during ontogeny, and the first isotype responding to antigenic stimulation, might be a leader of immune responsiveness. It has been demonstrated, using a murine model and classical model immunogens, that binding of complement onto the IgM antibody specific to immunogen enhances the humoral but not T-cell-dependent cellular responses [119]. IgM, similar to IgG3, as both are dominant natural antibody isotypes, regulates antibody response via complement and the complement receptors 1 and 2 (CR1/2) expressed on both B cells and follicular dendritic cells [120].

As previously presented, *Francisella* interacts directly with peritoneal murine B cells [18,19,20,121] and, as early as 12 h post infection, both germ-free and specific pathogen-free mice infected with *F. tularensis* produce infection-induced antibody clones reacting with *F. tularensis* proteins and having the character of natural antibodies. According to data in the literature, we characterized their functional profile as inducers of homeostasis restoration (Table 1). We also suggest that a phylogenetically stabilized defense mechanism utilizing early infection-induced antibody specificities can be activated not only to eliminate a pathogen but also to remove debris and molecular residues of infection-damaged self-cells [98].

### 4.3. Adaptive Immune Response: Antibody Functions

Antibody functions within the framework of immune mechanisms are specified by antibody molecule structure and by cellular antibody receptors, which ensure signal transduction and cellular response. Antibody molecules possess two functional domains. The antigen-binding fragment (Fab) confers antigen specificity. The Fc fragment (crystallizable fragment), generally known also as the constant fragment, drives other antibody functions. The Fc fragment variants, IgM, IgD, IgG, IgA, and IgE isotypes, have unique structural features that impact antibody function. Moreover, the IgG isotype has four subclasses (IgG1, IgG2, IgG3, and IgG4) and the IgA isotype has two subclasses (IgA1 and IgA2) [122]. The specific effector functions are triggered by binding of an antibody molecule to the appropriate receptor on the specific cell type, to which the antibody Fc domain binds [123,124]. These sensors include both classical FcRs and nonclassical C-type lectin receptors (CLRs), which are differentially expressed on immune cell subsets. The targeting of antibody function is further specified by posttranslational modification of FcRs [125]. Such a complex molecular basis of antibody function expression enables response to countless molecular and cellular targets that could compromise the integrity of the vertebrate organism. If we accept a simple classification of antibody functions, then antibodies are capable of neutralizing pathogen entry into the cell and its replication, neutralizing microbial virulence factors, ensuring antigen uptake, inducing antibody-mediated complement activation or antibody-dependent cellular cytotoxicity, and realizing antibody-dependent cellular phagocytosis, as well as antibody-mediated granulocyte degranulation and release of vasoactive mediators, chemoattractants, and cytokines. The dark side of their effects can involve antibody-dependent disease enhancement by altering the targeting of immune defense mechanisms [126].

Collectively, the data obtained from different *F. tularensis* infection models suggest that the antibodies against structural components of *F. tularensis* participate during both the innate and the adaptive phases of immune response. The functions of IgM isotype antibodies produced by B1a cells dominate during innate immune response [96,127], while functions of other isotypes seem to be needed during the adaptive phase of immune response against *F. tularensis*.

Results from a study by Furuya et al. showed that IgA^−/−^ mice were more susceptible than IgA^+/+^ mice to intranasal *F. tularensis* LVS infection that started in the second week post infection, despite developing higher levels of anti-LVS total, IgG, and IgM isotype antibodies in bronchoalveolar lavage [128]. Furthermore, comparison of vaccination via different routes (intradermal versus intranasal) suggests the involvement of IgA isotype antibodies in protective efficacy against the lethal effect of *F. tularensis* infection [129,130]. The protective efficacy of the IgA antibody isotype seems to involve a complex event encompassing, inter alia, cellular IFN-γ responses [128]. Experiments with *F. tularensis* lipopolysaccharide further demonstrated the role of IgA in IgG class switching after lipopolysaccharide vaccination. This modulatory effect, demonstrated on IgA^−/−^ mice, can be overcome by immunization with whole bacteria [131].

The IgG antibody isotype dominates during adaptive response to *F. tularensis* natural infection (i.e., from the second week after infection) in humans. Such a conclusion is supported by numerous studies, collectively discussed in a review by Maurin et al. [63]. Among the IgG subclasses, the IgG2 subclass prevailed, having been diagnosed in 92.9% of patients with serologically confirmed tularemia. The serum levels of IgG2 prevailed over those of IgG1 and IgG3, and the level of IgG4 was below the detection level [66]. The effects of IgG antibodies during the adaptive phase of immunity are produced mostly through receptors for their Fc fragments. The recognition of an FcR-targeted immunogen has a significant impact on the expression of immune defense mechanisms. Targeting of an inactivated *F. tularensis* live vaccine strain using mouse IgG2a anti-*F. tularensis* LPS mAb upon FcRs at mucosal sites (via intranasal immunization) enhances immunogen-specific IgA production and confers protection against subsequent infection in an IgA-dependent manner. Moreover, it enhances protection against the highly virulent SchuS4 strain of *F. tularensis*. Two types of FcRs, specifically FcgammaR and neonatal FcR, are crucial to this protection [132]. An effect similar to induction of a protective response as utilization of opsonized inactivated *F. tularensis* has been shown to occur following independent intranasal application of IgG2a anti-*F. tularensis* mAb and inactivated F. tularensis live vaccine strain [133]. This study design also showed an increase in protection against subsequent *F. tularensis* challenge that is FcR-dependent and requires a physical linkage between the monoclonal antibody and the inactivated *F. tularensis* immunogen [133].

If the role of anti-*F. tularensis* antibodies during the murine innate phase of immune response is mostly dependent on complement activation, then such a dependency has not been demonstrated using sera obtained during the adaptive phase of immune response or using convalescent sera. In this case, the protective effect of anti-*Francisella* antibodies has been shown to be independent of complement activation but the dependency on Fc receptors and phagocytosis was clearly demonstrated [134]. Studies on a combination of inactivated *F. tularensis* immunogen and immune sera or anti-*F. tularensis* mAb have revealed modulatory effects on adaptive immune mechanisms, starting from better protection against subsequent *F. tularensis* challenge, also in addition to including enhanced binding and internalization of inactivated *F. tularensis* by antigen-presenting cells through engagement of different FcR types, enhanced dendritic cell maturation, a prolonged time period through which Ag-presenting cells stimulate T cells, and modulated kinetics of inactivated *F. tularensis* immunogen transport from periphery to lymphoid tissues [132,133,134].

Antibacterial antibody-dependent cell-mediated cytotoxicity (ADCC) is a complex immune mechanism that can limit the proliferation of microbial pathogens inside the infected organism. Downregulation of microbial proliferation is effected through limitation of microbial proliferation inside the cells [135] or directly by killing free bacteria [136]. In contrast to complement, which also lyses targets but does not require any other cell, ADCC requires an effector cell that dominantly interacts with IgG antibodies bound to the surface of target cells. In this sense, the ADCC, utilizing the antibody as a critical component, is independent of the complement system. The typical effector cell in ADCC is an NK cell expressing an Fcγ receptor. Nevertheless, other cell types such as macrophages, neutrophils, or eosinophils [137], as well as other antibody isotypes, can also mediate antibacterial ADCC [138]. The NK cells having appropriate Fc receptors will bind to the corresponding antibody and will release proteins, such as perforin and granzymes, which cause lysis of the infected cell. One of the *Francisella* models, *F. novicida ΔfopC*, allowed documenting perforin-mediated inhibition of *F. tularensis* LVS replication in macrophages while identifying the NK cells as the critical cell type producing perforins [139]. This is one of the examples where anti-*F. tularensis* antibodies limit the proliferation of bacteria inside the body via the ADCC mechanism. Our unpublished data from the 1980s suggested the existence of ADCC-mediated limitation of *F. tularensis* proliferation in macrophages. The nonadherent spleen cells isolated from vaccinated mice 21 days post vaccination limited the number of *F. tularensis* in the cultures of in vivo infected peritoneal macrophages from naïve mice, but only when the sera from vaccinated mice were added to the cultures.

### 4.4. Protective Value of Anti-F. tularensis Antibodies

In early studies, the contribution of antibodies to host protection against *F. tularensis* appeared somewhat ambiguous. Passively transferred antibodies have been shown to confer protection against *F. tularensis* subsp. *holarctica*, including the LVS strains, but not against *F. tularensis* subsp. *tularensis* strains. This was documented mostly in the SCHU S4 infection model. At present, however, there remains controversy as to the contribution of B cells at a molecular level to protective immunity against *F. tularensis* infections. Both the production of cytokines affecting the functional profile of immunocompetent cells and the production of antibody molecules capable of modulating effector mechanisms of immunity come into consideration. Recently, it has become clear that specific antibodies against *F. tularensis* used for passive immunization are to some extent able to protect against lethal *F. tularensis* infection. The early studies using immune sera have already shown the indisputable protective effectiveness of the humoral components of the serum [140,141,142]. Later studies demonstrated that natural infection and vaccination induced long-lasting humoral and cell-mediated protective immunity; however, the humoral immunity gave protection only against strains having reduced virulence [143,144]. For this reason, interest has turned to the study of cell-mediated immunity mechanisms. At the turn of the millennium, several publications again stimulated interest to further study the humoral immunity in tularemia [145,146,147,148]. Different models of experimental tularemia gave rise to basic knowledge on the protective value of anti-*F. tularensis* antibodies. Passive transfer of immunity by specific antibodies against *F. tularensis* provided direct evidence that, due to the presence of pathogen-specific antibodies, this will benefit the host during infection caused by intracellular pathogens [149]. However, no protection was obtained in BALB/c mice against *F. tularensis* pulmonary infection by serum transfer from *F. tularensis* subsp. *holarctica* LVS-immune animals [150]. Passive transfer of immune sera also protected immunocompromised mice to some extent. Mice irradiated by sublethal gamma irradiation (3Gy) were protected against low lethal doses of the *F. tularensis* LVS, as well as against the original Soviet vaccine strain 15, which is more virulent for mice than is the LVS strain [151]. The protective effect of anti-*Francisella* antibodies has been shown to be dependent upon IFN-gamma production and on FcγR-mediated opsonophagocytosis and to be independent of complement activation. This might suggest a dominant role of ADCC in the protective efficacy against *F. tularensis* infection [134].

### 4.5. Secondary Immune Response: B Cells and Antibodies

Passive protection of sub-lethally irradiated mice against primary *F. tularensis* LVS infection also protected those mice that survived primary LVS infection against further secondary challenge with a highly virulent strain of *F. tularensis* subsp. *tularensis* SchuS4. Meanwhile, significantly fewer mice survived that were only LVS-vaccinated without an initial passive transfer of immunity [151,152]. Thus, the initial participation of humoral immune response to *F. tularensis* infection seems to play a substantial role in an effective protective secondary response. Parenteral intradermal and intraperitoneal *F. novicida* infections of wild-type mice or of B-cell knockout mice did not appreciably impact survival after subsequent lethal *F. novicida* challenge, thus demonstrating that B cells, if not serum antibodies, play a major role in controlling *F. novicida* infections in mice [153]. Studies have shown that, in constructing vaccines against tularemia, emphasis must also be given to inducing humoral immunity, which participates in a protective immune response. This is evidenced by a study with a combination vaccine containing ingredients that induce both humoral and cellular branches of immunity and protect against otherwise lethal intranasal and intradermal challenge with wild-type *F. tularensis* strains Schu S4 /type A/ and FSC 108 /type B/ [154].

## 5. Conclusions

The experimental data presented in the studies cited above clearly document that antibodies must be an integral part of induced immunity if it is to be truly protective. Protection against *F. tularensis* infection requires several discrete events. Early production of anti-*F. tularensis* antibody [98], which might be characterized as a booster of opsonophagocytosis of pathogens, is one of the key events in the induction of acquired immunity against bacterial pathogens. A second one is innate immune recognition and activation of antigen presentation. As antigen-presenting cells, dendritic cells, macrophages, and B cells are equipped with receptors recognizing pathogen-associated molecular patterns that mediate the innate immune recognition [4]. Another link in the functional chain of events providing protective immunity is the activation of effector cells that eliminate bacteria from the cells and tissues of the infected host. The last desirable protective event is the establishment of immune memory. Available data indicate that antibodies can initiate, regulate, or directly mediate these events.

There exist several approaches to harnessing the ability of antibodies to create effective and safe means of protection against highly virulent strains of *F. tularensis*. The simplest of these is a combination of passive and active immunization, which not only eliminates the side-effects of vaccination with live strains but also provides greater protection against subsequent virulent challenge. A second one consists of a generation of pathogen-specific IgM antibody clones produced by B1a B cells, which have been shown to induce protection while assuming the mutual interaction of cellular and humoral immune mechanisms [155], or in preparing antibody clones based on the knowledge of B-cell-activating epitopes on *F. tularensis* proteins [156]. Another, more sophisticated, approach is to construct a combined bacterial protein or whole bacterium with targeting (homing) components of eukaryotic molecules, such as Fc fragments of antibodies or the C3 component of complement, to provide directed opsonophagocytosis of tularemic antigens or of whole microbes and initiate the effective protective response (see, for example, Holland-Tummillo et al. [157]. However, we should once more repeat that, in relation to effective immunoprophylaxis of tularemia, we still have substantial gaps in our knowledge regarding the effective immune mechanisms, their collaboration and precise timing during innate and adaptive phases of immune response, and the bacterial molecular mechanisms interfering with induced immune responses. Thus, despite a number of very sophisticated studies on mutual host–pathogen interactions on cellular and molecular levels, it can be valuable to conduct further targeted studies regarding individual mechanisms of immunity enabling elimination of the microbe, including the role of B cells and their products.

## Figures and Tables

**Table 1 microorganisms-09-02136-t001:** Possible stages of infection-induced natural antibodies involvement in homeostasis restoration after infection. * nIg sAb—natural IgM soluble antibodies, ** IgRs—immunoglobulin receptors, *** i-nAb—infection-induced early (natural) antibodies.

	Effector	Target	Process
Stage 1	nIg sAb *	Microbial surface targets	Opsonization
Stage 2	Complement	nIg–antigen complex	Complement activation
Stage 3	C-Ig–Ag complex	CRs, IgRs ** and/or BCR	Cell–microbe interaction
Stage 4			B cell subset(s) activation
Stage 5			Production of i-nAb ***
Stage 6	i-nAb	Components of pathogen(s) and/or self-infection-damaged cells	Reinforced opsonization
Stage 7	i-nAb–(C)-Ag complex	Receptors of phagocytic and/or immunocompetent cells	Elimination and destruction of pathogens and damaged self-cells by phagocytes
Stage 8			Induction of adaptive immunity and immunological memory
Stage 9			Restored homeostasis

## Data Availability

Not applicable.

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
