# Peer review of "Francisella and Antibodies"

_microorganisms, 2021, doi:10.3390/microorganisms9102136_

Round 1

Reviewer 1 Report

In this review, the authors provided an overview of published studies related to the humoral responses against Francisella tularensis.  

The review appears to include citations of most of the studies performed on this subject. However, the written narrative in few points is hard to follows. The authors may consider some editing to make the reading more fluent.

Minor comments

A space between words is needed:  Line 35, between Francisellae and include. Line 38, between accomplish and their. Line 67, between tularensis and (35-37). Line 100, between assay and (ELISA)Line 158, between immunity and (96,97). Line 304, between induce and antibody. Line 321, between tularensis and infection. Line 343, between and and inactivated. Line 344, between (133) and This. Line 423, between (152) and Studies. Line 465, between Contributions: and K.K.

It appears that there is an extra space in: line 76, between (47-50): and snowshoe. Line 156, between LVS, and induced.

Across the manuscript, the authors use the word Francisellae, likely plural in Latin. Although it is correct, it is not often used, and it may induce confusion for the reader. The authors may consider change it.

Line 98: it is not clear what “occurred simultaneously” means. Are they produced at the same time after infection? Please re-phrase.

Lines 112-114: the sentence is not clear; please re-phrase the sentence to point out that cellular immunity persist longer that humoral.

Line 122: the word “produced” may be eliminated.

Line 126: the word “demonstrated” could be replaced with “identified”.

Line 327: It is not clear what “real” means. Is it referred to natural infection? Please re-phrase.

Line 343: F. tularensis is not in italic.

Line 353: mAB should be written mAb.

Line 377: nonadherent should be written non-adherent.

Line 379: in vivo is not in italic.

Lines 382 and 392: it is not clear what “original” means. Is it early studies? Novel studies? Please clarify.

Lines 433-434: the sentence should be re-phrased such as “Early production of anti-F. tularensis antibodies”.

Author Response

Comment 1: A space between words is needed:  Line 35, between Francisellae and include. Line 38, between accomplish and their. Line 67, between tularensis and (35-37). Line 100, between assay and (ELISA)Line 158, between immunity and (96,97). Line 304, between induce and antibody. Line 321, between tularensis and infection. Line 343, between and and inactivated. Line 344, between (133) and This. Line 423, between (152) and Studies. Line 465, between Contributions: and K.K.

Response 1: All spaces have been corrected in the text.

Comment 2: It appears that there is an extra space in: line 76, between (47-50): and snowshoe. Line 156, between LVS, and induced.

Response 2: The same as in the point above, all spaces have been corrected in the text.

Comment 3: Across the manuscript, the authors use the word Francisellae, likely plural in Latin. Although it is correct, it is not often used, and it may induce confusion for the reader. The authors may consider change it.

Response 3: Yes, we accepted, all the names in the plural of Francisellae have been converted to the singular form of Francisella

Comment 4: Line 98: it is not clear what “occurred simultaneously” means. Are they produced at the same time after infection? Please re-phrase.

Response 4: We modified the sentence as follows: The IgM, IgG, and IgA isotypes generally occurred simultaneously when the agglutination test was used for antibody titer evaluation during the first week after infection. Please see modified manuscript.

Comment 5: Lines 112-114: the sentence is not clear; please re-phrase the sentence to point out that cellular immunity persist longer that humoral.

Response 5: We modified the sentence as follows: However, the tests of cell-mediated immunity were more convincing than was agglutination test used for evaluation of humoral response (68), because cellular response persists longer than the antibody response.

Comment 6: Line 122: the word “produced” may be eliminated.

Response 6: The word was eliminated.

Comment 7: Line 126: the word “demonstrated” could be replaced with “identified”.

Response 7: The word was replaced.

Comment 8: Line 327: It is not clear what “real” means. Is it referred to natural infection? Please re-phrase.

Response 8: yes, we made a change (real versus natural).

Comment 9: Line 343: F. tularensis is not in italic.

Response 9: Format was changed.

Comment 10: Line 353: mAB should be written mAb.

Response 10: the abbreviation mAB was corrected.

Comment 11: Line 377: nonadherent should be written non-adherent.

Response 11: Nonadherent was corrected to non-adherent.

Comment 12: Line 379: in vivo is not in italic.

Response 12: In vivo was formated to italics.

Comment 13: Lines 382 and 392: it is not clear what “original” means. Is it early studies? Novel studies? Please clarify.

Response 13: At the line 382 the word was replaced. The sentence begining at the line 392 was modified to: The early studies using immune sera have already shown the indisputable protective effectiveness of the humoral components of the serum…

Comment 14: Lines 433-434: the sentence should be re-phrased such as “Early production of anti-F. tularensis antibodies”.

Response 14: The sentence has been modified as recommended.

Reviewer 2 Report

The paper by Kubelkova and Macela is a very interesting review. They provide a compreghensive review of recent knowledge and a multi-step model of immune response towards the pathogen.

I suggest to include a/some figure/s for illustration the diverse immune resposes. Moreover, I found one paper by Jacob et al (Outbreak of Tularemia in a Group of Hunters in Germany in 2018-Kinetics of Antibody and Cytokine Responses. Microorganisms. 2020 Oct 23;8(11):1645.), which has not been cited.

Author Response

Comment 1 :I suggest to include a/some figure/s for illustration the diverse immune resposes. Moreover, I found one paper by Jacob et al (Outbreak of Tularemia in a Group of Hunters in Germany in 2018-Kinetics of Antibody and Cytokine Responses. Microorganisms. 2020 Oct 23;8(11):1645.), which has not been cited.

Response 1: the article: Jacob et al .Outbreak of Tularemia in a Group of Hunters in Germany in 2018-Kinetics of Antibody and Cytokine Responses. Microorganisms. 2020 Oct 23; 8 (11): 1645, has been added to page 2 of the manuscript under number 157.

There are many schemes showing individual branches of the immune response against Francisella and they are generally valid. For this reason, we do not consider it necessary to insert a general scheme into this article intended only to illustrate the functionality of one branch of anti-Francisella immune response.